# Flexural Behavior on Damaged Steel Beams Strengthened with CFRP Sheets Subjected to Overloading

**DOI:** 10.3390/polym14071419

**Published:** 2022-03-30

**Authors:** Wenyu Hou, Fulong Huang, Kexin Zhang

**Affiliations:** School of Transportation Engineering, Shenyang Jianzhu University, Shenyang 110168, China; houwenyu61@163.com (W.H.); zhangkexin19892022@163.com (K.Z.)

**Keywords:** damaged steel beam, CFRP sheet, overloading, numerical analysis

## Abstract

This paper presents results of testing and numerical analyses of damaged steel beams strengthened with carbon-fiber reinforced polymer (CFRP) sheet and subjected to overloading. The test results showed that as cyclic times increase, the yield load and stiffness increase, but the ultimate bearing capacity decreases to a certain extent. Applying prestress could improve the ductility of the girder. The damage level greatly influenced the girders with overloading, and the increase in damage degree reduced the stiffness, yield load and ultimate load. The numerical calculation showed that the yield load would decrease by 0.4–0.8 kN when the cyclic times increase by 100 times. The yield load would increase by 1–2 kN and the ultimate bearing capacity would decrease by 0.6–1.2 kN when the overloading amplitude increased by 0.02 Pu. The increase in damage degree would obviously reduce the yield load and ultimate bearing capacity of the steel beam after overloading. The yield load and ultimate bearing capacity could be increased by approximately 3 kN when the CFRP sheet thickness increases by 0.05 mm.

## 1. Introduction

Steel structures may be damaged by environmental erosion, natural disasters, improper operation, and load and human factors during long-term use. [1,2]. The damage reduces the bearing capacity and durability of the structure and may even lead to failure. Carbon-fiber reinforced polymer (CFRP) sheets have unique material and mechanical properties such as low weight, high strength and stiffness and good durability. They can be epoxy-bonded to the tension face of damaged steel members to restore or enhance their ultimate bearing capacities [3,4]. After reinforcement, some steel structures are still subjected to overloading in which the service load is greater than 60% of the ultimate load of the structure. Long-term overloading reduces the bearing capacity and ductility of the structure. Some experts have studied the fatigue properties of steel structures. Zheng et al. [5] conducted an experimental study on the fatigue performance of CFRP-reinforced steel beams. A four-point bending method was used to conduct fatigue tests on H-shaped steel beams with a 1.4 m span and 0.6 m adjacent load point length, with stress amplitude ranging from 85 MPa to 156 MPa and a stress ratio of 0.2. Furthermore, three girders were control girders, two girders were double-sided and strengthened with CFRP with a length of 0.4 m, another three girders were double-sided and strengthened with CFRP with a length of 1 m and one girder was double-sided and strengthened with CFRP with a length of 1 m at the pre-damaged part of the lower flange of the steel beam. The results showed that the fatigue life of the steel beams strengthened with CFRP could be increased to 2.98 times that of the original, and the fatigue life of the steel beams strengthened with CFRP could be restored to an undamaged state. Colombi et al. [6] studied the mechanical fatigue properties of cracked steel beams strengthened by CFRP plates. The test results showed that using a CFRP plate at a crack could reduce crack development and prolong fatigue life. Stress concentration of the CFRP plate occurred at the cracks, and the probability of the CFRP plate debonding at the crack was greater. Zhang and Yao [7] studied the flexural bearing capacity of CFRP-reinforced steel beams and CFRP-reinforced steel members under cyclic loading for engineering application and further research reference. The results showed that the flexural bearing capacity and stiffness of steel beams strengthened with CFRP could be significantly improved. The thickness of the CFRP plate was the key to the increased range. Increasing the length of the CFRP plate could prevent premature debonding failure. Colombi and Fava [8] used an experiment, numerical simulation and theory to study the mechanical properties of damaged steel beams strengthened with CFRP plates under fatigue loading. The results showed that CFRP plates could effectively inhibit the development of cracks and prolong the fatigue life of steel beams. A debonding area existed between the CFRP plate and steel beam at the damaged position, which had a certain influence on the reinforcement effect. Deng et al. [9] used linear elastic fracture mechanics to study the fatigue life of steel beams strengthened by CFRP plates; the results showed that the established formula for calculating the fatigue life of these steel beams had strong practicability and was in good agreement with the test data. Aljabar et al. [10] studied the fatigue crack growth and fatigue life of a rectangular steel plate with a central crack strengthened by CFRP under tensile cyclic loading and studied the influence of different crack angles; a correction coefficient of fatigue life with type I (open) and type II (slip) mixed crack characteristics was proposed. Yu et al. [11] used numerical analysis, testing and fracture mechanics analysis to study the fatigue crack propagation and fatigue life of the rectangular steel plate with a through crack at the edge strengthened with CFRP laminate under cycle loading. The results proposed the fatigue life calculation method or the value of the correlation coefficient and showed that CFRP could significantly reduce the crack propagation rate. Moreover, double-sided CFRP reinforcement was better than single-sided CFRP reinforcement for raising fatigue life. Yu and Wu [12] used two methods to reinforce six damaged steel beams; one was only pasting CFRP plates, and the other was pasting and anchoring CFRP plates. Fatigue tests were conducted on all strengthened steel beams, and the results showed that additional anchoring of CFRP plates could delay the development of the damaged position of the steel beam and prolong fatigue life. The effects of bond slip and bond length on mechanical fatigue properties were small. Yang et al. [13] used cyclic loading to analyze the binding point of CFRP and steel, as well as its strength, bearing capacity and bond slip curve. The results showed a critical value; this value would not cause detectable damage. When the number of cycles exceeded this critical value, the nodal region near the loading end would degenerate; the relationship between damage and slip was established using experimental data, and a preliminary estimate of the fatigue limit was proposed. Ye et al. [14] studied the use of an unbonded prestressed CFRP plate to reinforce the damaged steel beam by using a large-scale fatigue model and analyzed the effect of different prestress levels on the fatigue life of the damaged steel beam. The results showed that applying prestress could decrease the residual deflection and crack propagation rate. The fatigue life of the damaged steel beams strengthened with the highest prestress level was increased by at least eight times and the fatigue strength of the notched steel beams strengthened with prestressed CFRP plates increased from 51 MPa to 75 MPa. Hu and Feng [15] presented a design method for a CFRP-reinforced damaged steel structure and developed the design program. The results showed that CFRP reinforcement can improve usable life under a certain stress range and the allowable stress range under the premise of achieving the target usable life.

To sum up, studies on the fatigue behavior of CFRP-reinforced steel beams are mainly about fatigue life, fatigue failure mechanism, bearing capacity, stiffness and other attributes. The effects of prestressed CFRP, CFRP thickness, cyclic times, cycle loading and other parameters on fatigue performance are analyzed by testing, finite element calculation and theoretical calculation. However, few studies have been conducted on the mechanical properties of the structure during overloading. In this study, the mechanical overloading behavior of damaged steel beams strengthened with CFRP sheets is studied by testing and finite element simulation, and the overloading cyclic times, overload amplitude, damage degree and other parameters are analyzed.

## 2. Experimental Program

The test used typical Chinese Standard steel I14A (Tianjin Jiangtian Section Steel Co. LTD, Tianjin, China) to be the steel beam. The depth of the steel beam was 140 mm, the width of the flange was 80 mm, the thickness of the flange and web were 9.1 mm and 5.5 mm, respectively and the area of section was 2150 mm^2^. There were three damaged steel beams in the test and the steel beams had two damaged levels of 50% and 100% loss of tensile flange, as shown in Figure 1. The 50% damage level meant that 50% of the length was cut off on each side of the tensile flange and 100% damage level meant that the tensile flange was completely cut off, only leaving the web of the steel beam. The tensile test results indicated that the yield strength and tensile strength of I-shaped steel were 256 MPa and 423 MPa, respectively. This test used a prestressed CFRP sheet and a CFRP sheet (Liaoning provincial building design and research institute, Shenyang, China) to strengthen steel beams. The thickness of the CFRP sheet was 0.167 mm, the width was 50 mm and the length was 1300 mm. Average tensile strength of the CFRP sheet was 3456 MPa and the elastic modulus was 258 GPa. The prestress was applied to the CFRP sheets by using a self-made stretching bed as shown in Figure 2. The implementation method involved fixing CFRP sheets on another larger I-beam by two steel slabs with four bolts at the end of the stretching bed and four screws which allowed upward and downward movement; and applying prestress to the CFRP sheets by raising the supports horizontally. Thereafter, the prestressed CFRP sheet and the CFRP sheet were pasted on the bottom of the tension flange U-shape hoops were used to ensure that the CFRP sheet could be anchored to the steel beams. The detailed parameters of the steel beams are provided in Table 1. The loading device for the test is shown in Figure 3. The hydraulic jack was used to apply the load and the spreader beam was used to divide the load equally between the two supports. In the overloading process, the minimum load (*P*_min_), maximum load (*P*_max_) and cyclic time were set. Then, the steel beams were cycled in this range. One cycle lasted approximately 6 min, as shown in Figure 4. After overloading, an entire load was applied to the steel beams until they were disrupted.

## 3. Test Results

### 3.1. Load-Deflection Curves of Cyclic Stage

Figure 5 shows the load-deflection curves at the cyclic stage. In the unloading process of CSB1, CSB2 and CSB3 after 100 cycles, the curves do not directly decline according to the original curves. This condition shows that the overload amplitude exceeded the critical value of the elastic stage of the steel beam, and all of them were in the elastic-plastic stage. As the notches of the three girders were small, the load-deflection curves are relatively smooth.

### 3.2. Failure Mode

The steel beams strengthened with a prestressed CFRP sheet or a CFRP sheet after overloading displays two distinct failure modes: buckling of the top flange and rupture of the CFRP sheet. At the initial stage of loading, the displacement of the girders was very small. When the load reached 57.2%Pu of CSB1, 55.2%Pu of CSB2 and 54.7%Pu of CSB3, the tensile flanges of the steel beams yielded. When the load reached 83.3%Pu of CSB1, 66.1%Pu of CSB2 and 83.7%Pu of CSB3, the compressive flanges of the steel beams yielded. When the load reached 98.4%Pu of CSB1, 94.9%Pu of CSB2 and 99.3%Pu of CSB3, the CFRP sheet began to make some noise. Then, the steel beams were destroyed. The failure of CSB1 indicated buckling of the top flange. The CFRP sheet rupture occurred in CSB2 and CSB3. The failure modes are shown in Figure 6.

### 3.3. Load-Deflection Curves before and after Overloading

The first and last time of load-deflection curves of CSB1, CSB2 and CSB3 are shown in Figure 7. The figure shows that the stiffness of the girders improved after 100 cycles. At the same load, the stiffness of the last loading of CSB1, CSB2 and CSB3 are respectively 50%, 12.6% and 49.2% higher than that of the first loading. The reason is that after a certain number of cycles, the steel had undergone cold hardening, and the stiffness of the girder was improved. The greater the corrosion degree, the lesser is the improvement of the girder stiffness after overloading.

### 3.4. Analysis of Parameters

#### 3.4.1. Prestress

The load-deflection curves of CSB1 and CSB3 are shown in Figure 8. CSB1 is the girder only strengthened with the CFRP sheet and CSB3 is the girder strengthened with the prestressed CFRP sheet. Before the yield load, the stiffness changes of the two girders were basically same. With the increase in load, the yield load and ultimate load of CSB4 increased by 2% and 1.6% more than that of CSB1, but the ductility of CSB3 was significantly higher than that of CSB1. This condition of applying prestress had no significant effect on the yield and ultimate bearing capacity of the girders with overloading, but it did improve the ductility of the girder.

#### 3.4.2. Damage Degree

The load-deflection curves of CSB2 and CSB3 are shown in Figure 9. CSB2 is the girder with 100% a damage degree and CSB3 is the girder with a 50% damage degree. At the initial stage of loading, the stiffness of CSB2 was smaller than that of CSB4. Before the end of the linear elastic section, the stiffness of CSB3 was 83.1% higher than that of CSB2. With the increase in load, the yield load and ultimate load of CSB3 were respectively 92.1% and 59.8% higher than that of CSB2. This result shows that the damage level strongly influences the girders with overloading. The increase of the damage degree obviously reduces the stiffness, yield load and ultimate bearing capacity.

## 4. Numerical Analysis

### 4.1. Constitutive Relation

In order to verify the accuracy of the theoretical calculation, finite element calculation software Abaqus was used in this study. The constitutive relation between the steel beam and the CFRP sheet in the Abaqus model was described by the plastic analysis model. Figure 10 and Figure 11 show the constitutive relationship between steel and the CFRP sheet [16,17,18], and the Poisson’s ratio of steel is 0.3. CFRP sheet is an orthotropic material with no strength in the fiber’s vertical direction. The stress–strain curve of the CFRP sheet was taken as ideal elasticity, and the elastic modulus of the CFRP sheet was consistent with the experimental value. Geometry and loading arrangements of the model were adopted in accordance with the experimentally tested beams. A three-dimensional element (C3D8R) was created for the steel beam and shell-type element (SR4) for the CFRP sheet. A structural grid method was used to discretize the girder. The mesh sizes of the steel beam and CFRP sheet was similar. A steel plate with a great degree of stiffness was set at the place of symmetrical concentrated load as support. The steel beam and the steel plate with a great degree of stiffness was bound by ‘Tie’. And ‘Tie’ was also used to constrain the tensile flange of the steel beam and the CFRP sheet. The load was simulated by applying displacement on the top of the beam, and the nonlinear equations were solved via incremental iteration method. The falling temperature method was used to apply prestressing to the CFRP sheet. The concrete implementation method was used to cool the CFRP sheet by applying a temperature load and making it shrink, so that the whole CFRP sheet was prestressed.

### 4.2. Calculation Results

#### 4.2.1. Stress Analysis

The stress diagrams of the steel beam and CFRP sheet are shown in Figure 12 and Figure 13. This figure indicates that the steel beam and CFRP sheet both had residual stress after overloading. The stress concentration at the notch of the steel beam appeared after overloading. The residual stress of the CFRP sheet was mainly concentrated in the mid-span. After the last loading, the final stress of the steel beam with overloading was smaller than that of only static loading. This condition shows that overloading has a certain influence on the final stress of the steel beam. The steel beam with overloading broke earlier than that of the static loading. The final stress of the CFRP sheet with overloading was smaller than that of the static loading. This condition indicates that overloading can reduce the utilization rate of the CFRP sheet.

Figure 14 shows the stress of the steel beam and CFRP sheet along the length. It shows that the tensile and compressive stresses of the steel beam are both the largest in the mid-span and increase suddenly near the mid-span. This result indicates that overloading could cause stress concentration in the mid-span of the steel beam. The stress of the CFRP sheet is also the largest in the mid-span, and the stress in other parts is similar. When the steel beam breaks, the stress concentration occurs because of the notch of the lower flange, and the stress concentration also occurs at the loading point of the steel beam. The tensile stress at the loading point is the largest, followed by that in the mid-span, and the stress in the compression zone is similar. The stress on the steel beam and CFRP sheet is lesser than that on the static loading girder. This result shows that overloading can reduce the stress of the steel beam and CFRP sheet.

#### 4.2.2. Deflection Analysis

Figure 15 shows the deflection distribution of the damaged steel beams strengthened with CFRP sheets subjected to overloading. This result shows that overloading could cause residual deflection. The final deflection is smaller than that of the static loading, indicating that the girder with overloading cannot reach the maximum deflection.

#### 4.2.3. Comparison between Test Results and Finite Element Calculation Results

Figure 16 shows a comparison between the test results and the finite element calculation results. The finite element calculating results are in good agreement with the test results, which indicate that the calculation model of the damaged steel beams strengthened with CFRP sheets subjected to overloading is correct.

#### 4.2.4. Parameter Analysis

Cyclic times

The load-deflection and stress along the length curves of girders with different cycles of 100–1000 times and only static loading are shown in Figure 17. The stiffness of the girders with cycles of 100–1000 times is greater than that of the girder with only static loading, but the difference in stiffness between the girders with cycles of 100–1000 times is very small. The yield loads of the girders with cycles of 100–1000 times are 13.8%, 12.9%, 12.1%, 11.2%, 10%, 9%, 8.1%, 7.3% and 6.4% higher than that of the girder with only static loading. However, the ultimate loads are lower than that of the girder with only static loading, and the ultimate loads of the girders with cycles of 100–1000 times do not differ considerably. This result shows that overloading can increase the yield loads of the girders, but the rate of increment decreases with the increase in the number of cycles. The yield load decreases by 0.4–0.8 kN for every 100 cycles. Overloading can reduce the ultimate bearing capacity of the girders, and the number of cycles has a minimal effect on the ultimate bearing capacity.

The tensile and compressive stresses in the bending section of the steel beam with cycles of 100–1000 times are smaller than that of the girder with only static loading. The tensile and compressive stresses in the bending and shear sections are slightly greater than that of the girder with only static loading. With the increase in the number of cycles, the tensile stress of the steel beam decreases. The tensile stress in the mid-span of the steel beam decreases by approximately 0.6 N/mm^2^ for every 100 cycles. The compressive stress in the mid-span of the steel beam changes minimally with the increase in the number of cycles, but the compressive stress in the bending and shear section grows with the increase in the number of cycles. The compressive stress in the bending and shear section increases by approximately 4.3 N/mm^2^ for every 100 cycles. This condition shows that the increase in the number of cycles can cause stress redistribution of the steel beam. The stress on the CFRP sheet of the girders with 100–1000 cycles is lesser than that of the girder with only static loading. The stress on the CFRP sheet decreases with the increase in the number of cycles, indicating that the increase in the number of cycles would reduce the utilization rate of the CFRP sheet to a certain extent.

Overloading amplitude

The load-deflection and stress along the length curves of the girders with overloading amplitudes of 0.6–0.76 Pu and only static loading are reported in Figure 18. The stiffness of the girders with the overloading amplitude of 0.6–0.76 Pu is greater than that of the girder with only static loading. Before the end of the linear elastic section, the stiffness of the girders with the overloading amplitude of 0.6–0.76 Pu respectively increased by 6.3%, 6.6%, 8.4%, 11.4%, 14.3%, 15.7%, 16.4%, 18.6% and 20% compared to that of the girder with only static loading. The yield loads increased by 16%, 18%, 21%, 22.9%, 25.1%, 26%, 27.2%, 29, and 30.8% compared to that of the girder with only static loading. However, the ultimate loads decreased by 2%, 2.4%, 2.9%, 3.3%, 3.9%, 4.6%, 5.8%, 6.8% and 7.9% compared to that of the girder with only static loading. It shows that the stiffness and yield load increase with the increase of overloading amplitude. The yield loads increase by 1–2 kN with an increase of 0.02 Pu, but the ultimate load decreases by 0.6–1.2 kN with an increase of 0.02 Pu. When the overloading amplitude is greater than 0.77 Pu, the deflection of the steel beam in the overloading stage is greater than the maximum deflection of the girder with only static loading, which means that the steel beam is damaged in the overloading stage.

The tensile stress in the bending section of the overloading steel beam is lesser than that with only static loading. However, the tensile stress in the bending and shear section is greater than that with only static loading, especially in the end part of the girder. When the overloading amplitude is 0.6–0.74 Pu, the tensile stress in the mid-span does not change considerably. When the overloading amplitude is over 0.74 Pu, the tensile stress in the mid-span decreases greatly. When the overloading amplitude is 0.6–0.74 Pu, the tensile stress at the loading point shows minimal change. When the overload amplitude is over 0.7 Pu, the tensile stress at the loading point decreases gradually. With the increase of the overloading amplitude, the stress of the CFRP sheet decreases, but when the amplitude is greater than 0.7 Pu, the reduction increases. This condition shows that the greater the overloading amplitude is, the greater the influence on the stress of the steel beam and CFRP sheet.

Damage degree

The load-deflection and stress along the length curves of girders with different damage degrees of 10%–100% are shown in Figure 19. At the initial stage of loading, the stiffness of the girders with a high damage degree is lower than that of the girders with a low damage degree. The yield load of the girders with a damage degree of 10%–90% respectively increased by 84%, 76%, 67%, 59%, 45.6%, 33.9%, 23.7%, 13.5% and 3.4% compared with that of girders with a damage degree of 100%. The ultimate loads increased by 84%, 76%, 67%, 59%, 50.7%, 38.7%, 28.1%, 17.6% and 8.1% compared with that of the girders with a damage degree of 100%. These results show that the increase of the damage degree significantly reduces the yield load and ultimate load. When the damage degree increases by 10%, the yield load and ultimate load decrease by approximately 8 kN.

The tensile stress in the mid-span of the steel beam is the largest, followed by the loading point, and the compressive stress in the mid-span is similar to the loading point. The tensile and compressive stresses of the damage degree of 10%–70% are similar. When the damage degree reaches 80%, the tensile and compressive stresses decrease obviously, but the tensile stress in the mid-span is similar to other damage degrees. The stress on the CFRP sheet in the mid-span is the largest, followed by the loading point. When the damage degree is 10%, the stress on the CFRP sheet is the largest. As the damage degree increases, the stress on the CFRP sheet decreases gradually. This result shows that when the damage degree is 10%, the utilization rate of the CFRP sheet is the highest. With the increase in damage degree, the utilization rate of the CFRP sheet decreases.

Thickness of CFRP sheet

The load-deflection and stress along the length curves of the girders with different thicknesses of the CFRP sheet in the range of 0.1–0.55 mm are shown in Figure 20. At the initial stage of loading, the stiffness of the girders with thinner CFRP sheets is lesser than that of the girders with thicker CFRP sheets. The yield loads of the girders with thicknesses of 0.15–0.55 mm respectively increase by 3.2%, 6.5%, 8.6%, 11.8%, 14.8%, 17.1%, 20%, 22.2% and 24.4% compared with the girders with a thickness of 0.1 mm. The ultimate loads increase by 3.2%, 6.4%, 8.6%, 11.5%, 14.6%, 17%, 19.8%, 22% and 24%. This condition shows that the increase in the thickness of the CFRP sheet increases the yield load and ultimate load. Specifically, the yield load and ultimate load rise by approximately 3 kN when the thickness of the CFRP sheet increases by 0.05 mm.

The tensile and compressive stresses on the steel beam are the largest in the mid-span and at the loading point. The changes in the tensile and compressive stresses with the thickness of the CFRP sheet are not obvious. The stress on the CFRP sheet is the greatest in the mid-span, followed by the loading point, and the stress on the CFRP sheet decreases with the increase of the thickness. This result shows that the change in the thickness of the CFRP sheet has a minimal effect on the tensile stress and compressive stress of the steel beam and stress of the CFRP sheet.

## 5. Conclusions

The test results showed that the steel beams repaired with prestressed CFRP sheets or CFRP sheets after overloading displayed two distinct failure modes: buckling of the top flange and rupture of the CFRP sheet. The failure mode of the damaged steel beam with only static loading is partial debonding of the CFRP sheet and partial rupture. Applying prestress has no significant effect on the yield and ultimate bearing capacity of the girders with overloading, but it can improve the ductility of the girder. The damage level strongly influences the girders with overloading. The increase of the damage degree obviously reduces the stiffness, yield load and ultimate bearing capacity.By comparing the numerical and test results, we found an agreement regardless of whether the steel beams were subjected to overloading or only static loading. The load-deflection curve was nonlinear and included three phases: elastic, elastic–plastic and plastic. Overloading can cause residual deflection and the girder with overloading cannot reach the maximum deflection as the static loading girder. Overloading can reduce the stress of the steel beam and CFRP sheet.The numerical analysis results showed that the yield load decreased by 0.4–0.8 kN when the cyclic times increased by 100 times. The increase in the number of cycles can cause the stress redistribution of the steel beam but would reduce the utilization rate of the CFRP sheet to a certain extent. When the overloading amplitude increased by 0.02 Pu, the yield load would increase by 1–2 kN and the ultimate bearing capacity would decrease by 0.6–1.2 kN. With the increase of the overloading amplitude, the stress of the CFRP sheet decreases, but when the amplitude is greater than 0.7 Pu, the reduction increases. The greater the overloading amplitude is, the greater the influence on the stress of the steel beam and CFRP sheet. The increase in damage degree obviously reduced the yield load and ultimate bearing capacity of the steel beam after overloading. When the damage degree is 10%, the utilization rate of the CFRP sheet is the highest. With the increase in damage degree, the utilization rate of the CFRP sheet decreases. The yield load and ultimate bearing capacity of the damaged steel beam could be increased by approximately 3 kN when the thickness of the CFRP sheet is increased by 0.05 mm. The change in the thickness of the CFRP sheet has a minimal effect on the tensile stress and compressive stress of the steel beam and stress of the CFRP sheet.

## Figures and Tables

**Figure 1 polymers-14-01419-f001:**
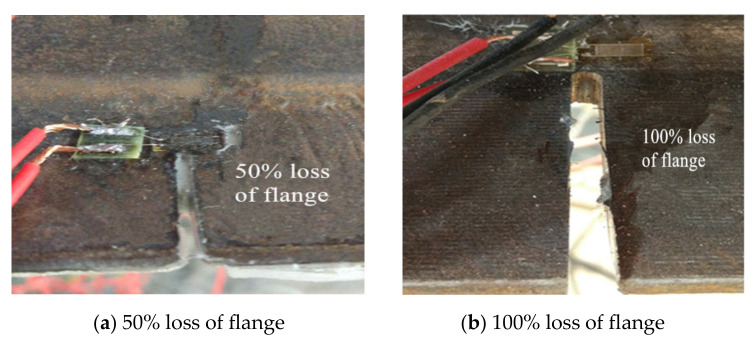
Damaged model of steel beams.

**Figure 2 polymers-14-01419-f002:**
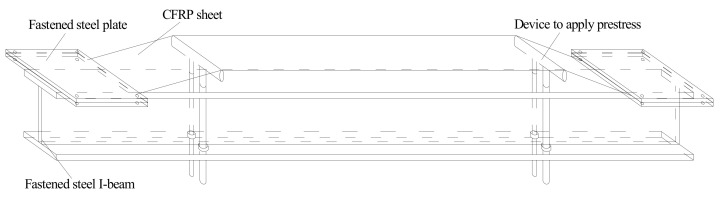
Device for applying prestress to CFRP sheet.

**Figure 3 polymers-14-01419-f003:**
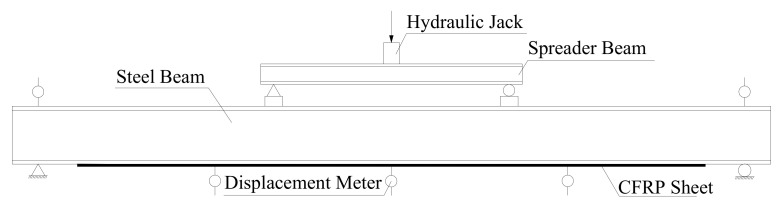
Loading device for testing.

**Figure 4 polymers-14-01419-f004:**
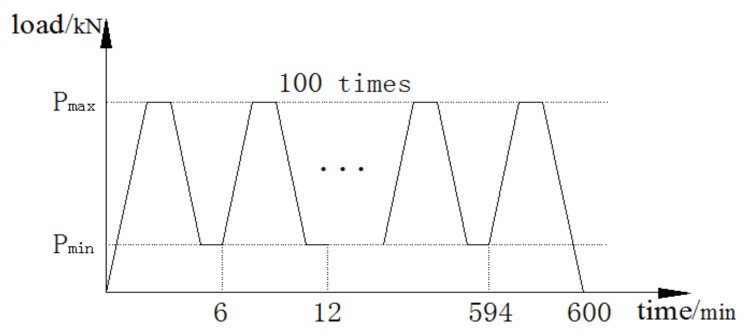
Overloading process.

**Figure 5 polymers-14-01419-f005:**
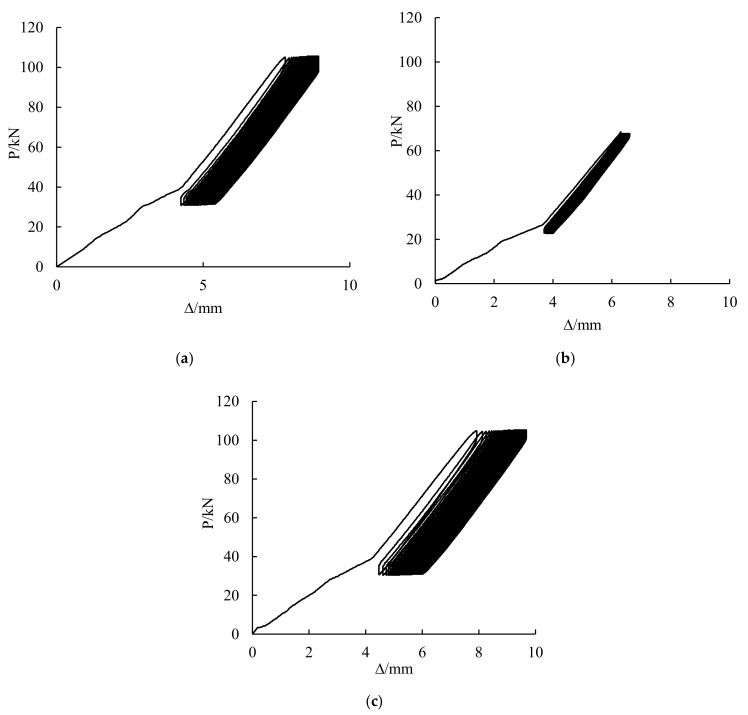
Load-deflection relationship of girders with 100 cycles of CSB1 (**a**), CSB2 (**b**) and CSB3 (**c**).

**Figure 6 polymers-14-01419-f006:**
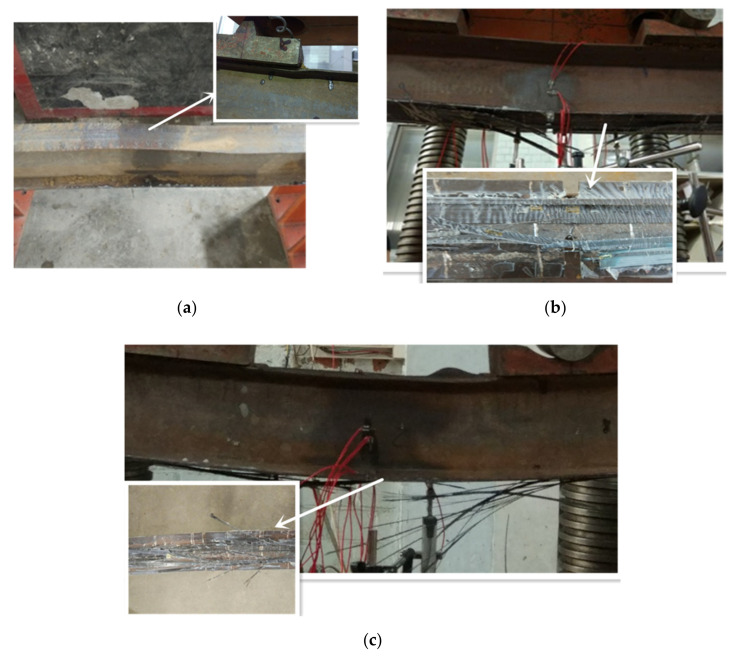
Failure modes of CSB1 (**a**), CSB2 (**b**) and CSB3 (**c**).

**Figure 7 polymers-14-01419-f007:**
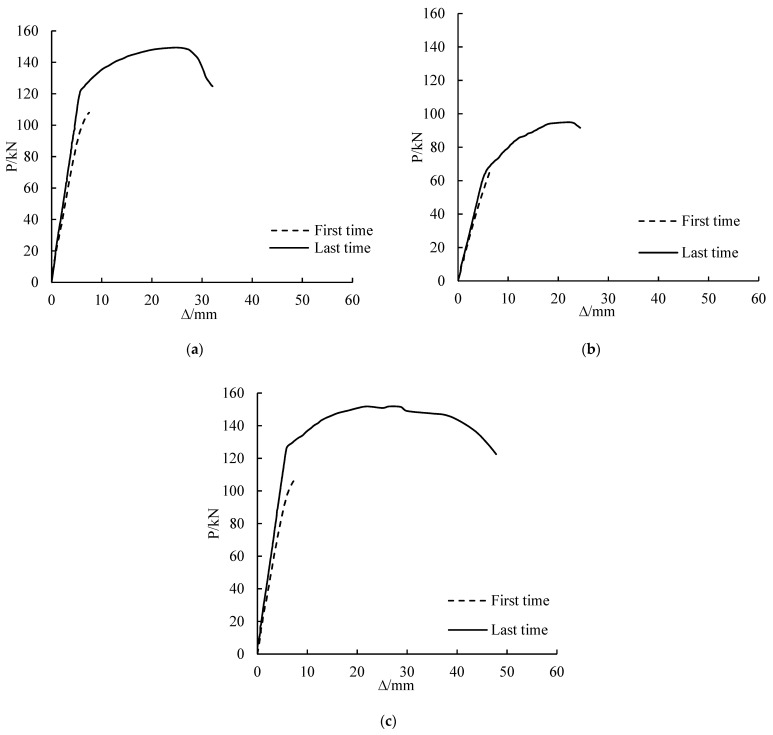
Load-deflection curves before and after overloading of CSB1 (**a**), CSB2 (**b**) and CSB3 (**c**).

**Figure 8 polymers-14-01419-f008:**
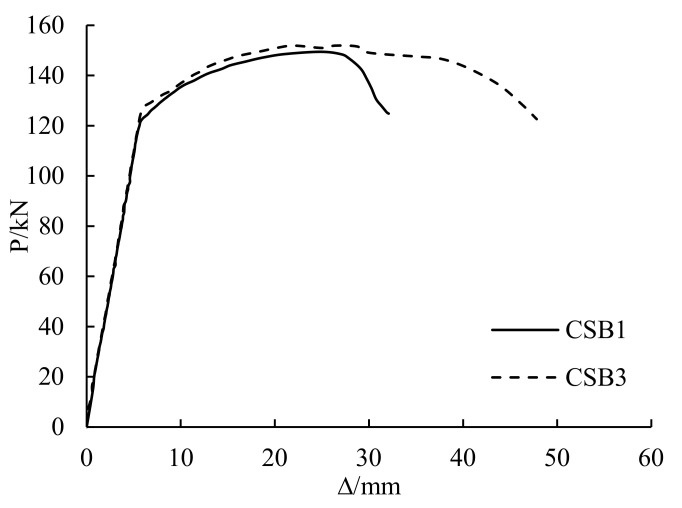
Influence of prestress.

**Figure 9 polymers-14-01419-f009:**
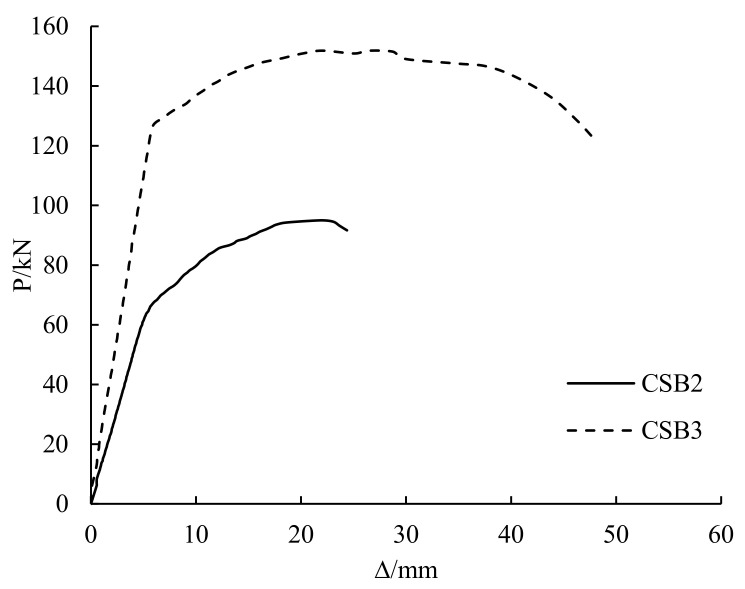
Influence of damage degree.

**Figure 10 polymers-14-01419-f010:**
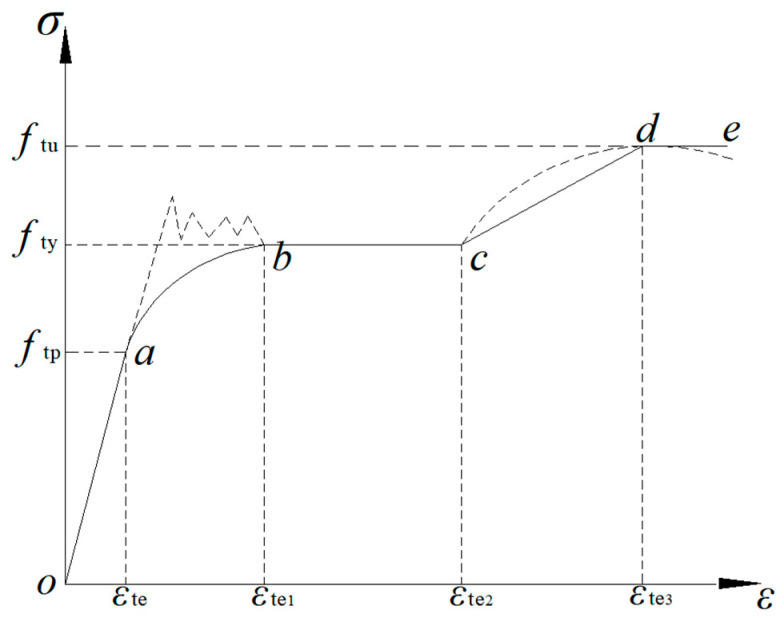
Stress–strain curve of steel.

**Figure 11 polymers-14-01419-f011:**
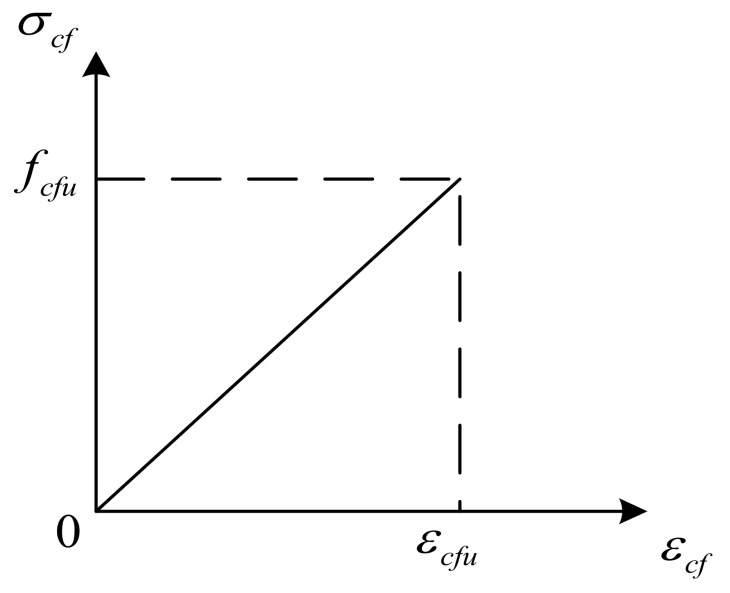
Stress-strain curve of CFRP sheet.

**Figure 12 polymers-14-01419-f012:**
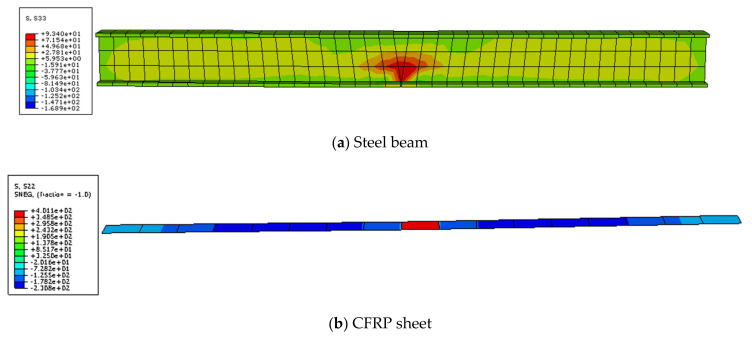
Stress image of steel beam (**a**) and CFRP sheet (**b**) after overloading.

**Figure 13 polymers-14-01419-f013:**
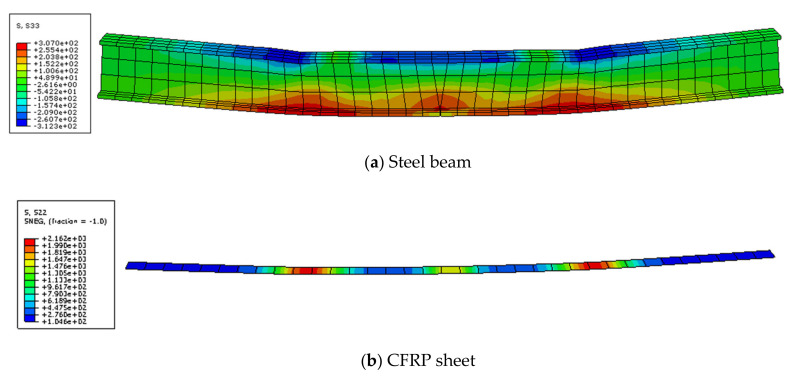
Stress image of steel beam (**a**) and CFRP sheet (**b**) after last loading.

**Figure 14 polymers-14-01419-f014:**
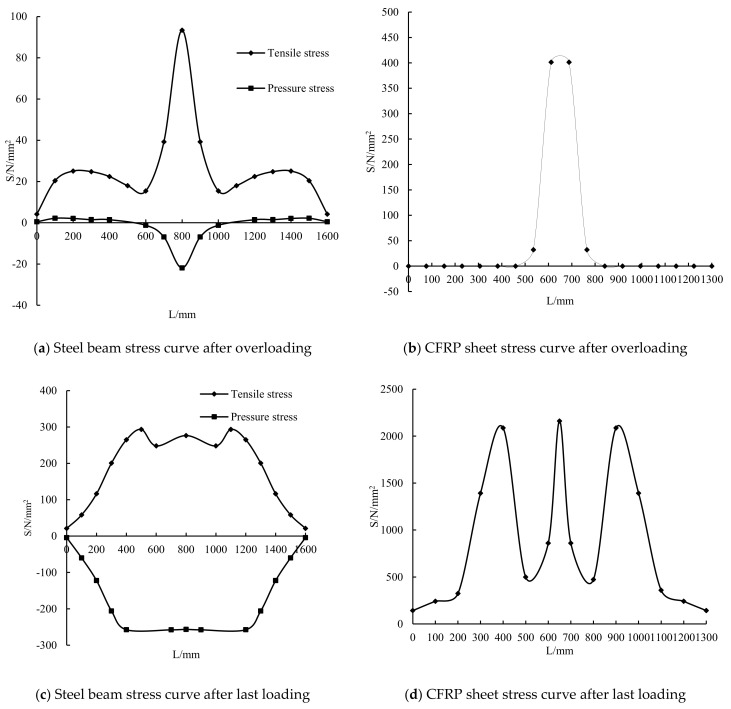
Stress distribution of steel beam and CFRP sheet along length.

**Figure 15 polymers-14-01419-f015:**
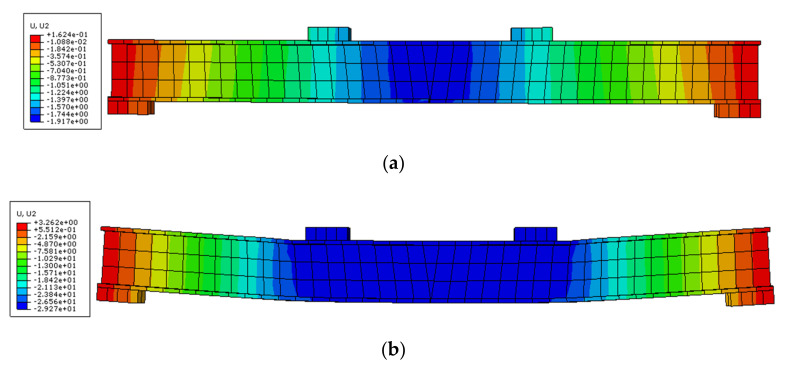
Displacement distributions of damaged steel beam strengthened CFRP sheet after overloading (**a**) and after last loading (**b**).

**Figure 16 polymers-14-01419-f016:**
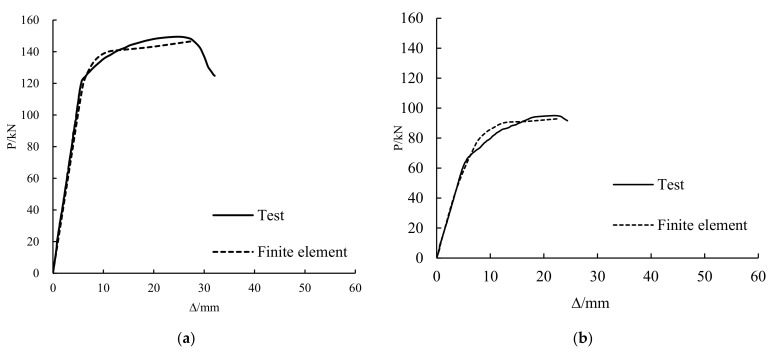
Comparison between test and finite element calculation of CSB1 (**a**), CSB2 (**b**) and CSB3 (**c**).

**Figure 17 polymers-14-01419-f017:**
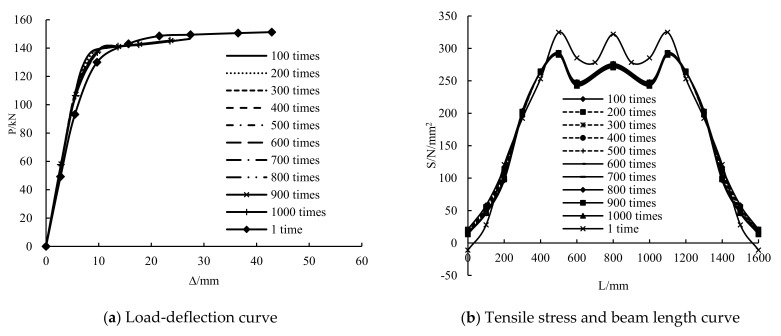
Influence of cyclic times.

**Figure 18 polymers-14-01419-f018:**
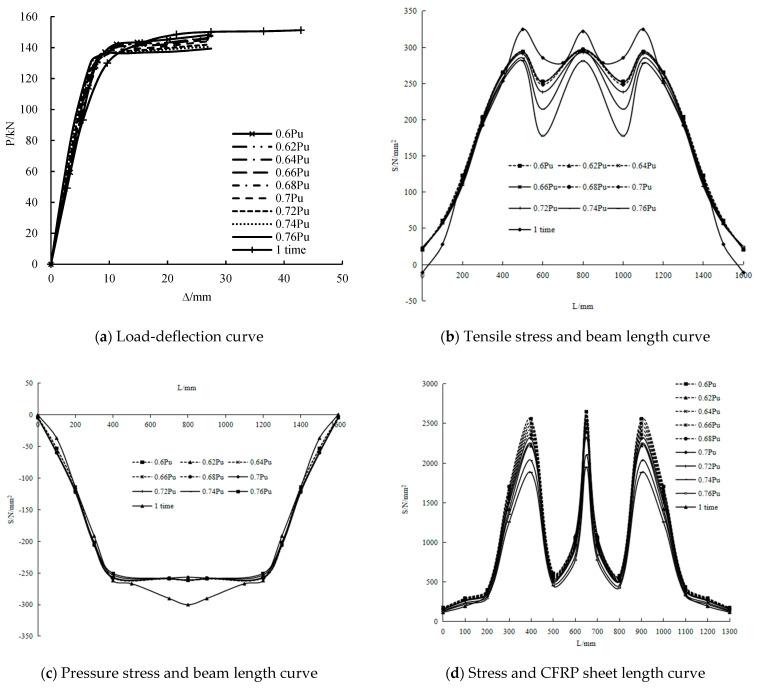
Influence of overloading amplitude.

**Figure 19 polymers-14-01419-f019:**
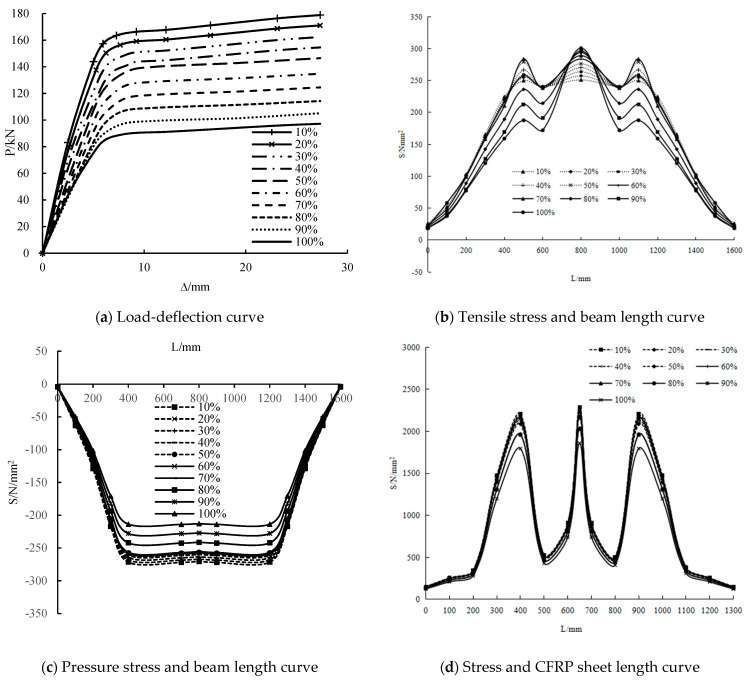
Influence of corrosion.

**Figure 20 polymers-14-01419-f020:**
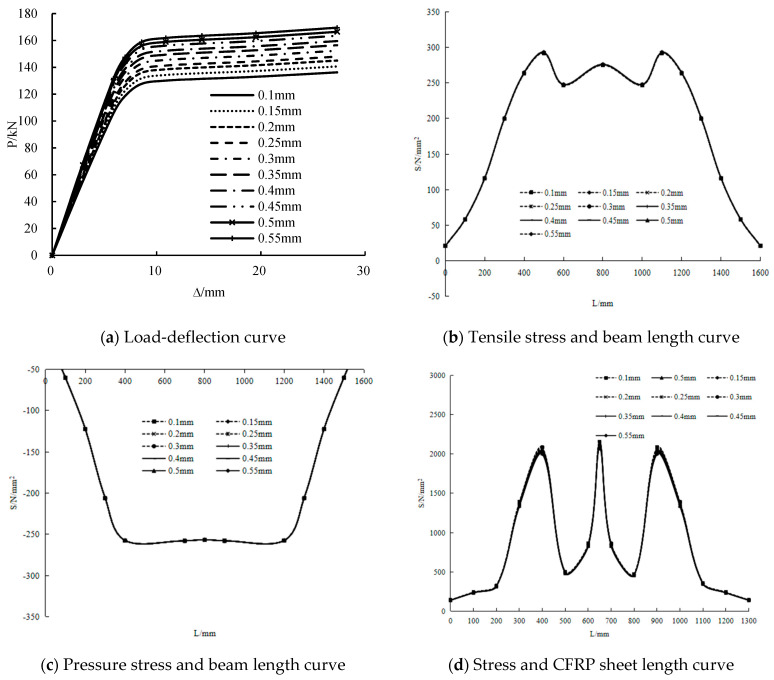
Influence of CFRP sheet thickness.

**Table 1 polymers-14-01419-t001:** Test matrix and results.

Girder Number	Prestressed Degree/ (%Pu)	Damaged Level/%	Type of CFRP	Overloading Amplitude	Overloading Number	Yield Load of Bottom Flange Pt(kN)/(%Pu)	Yield Load of Top Flange Py (kN)/(%Pu)	Ultimate Load Pu (kN)	Failure Modes
CSB1	0	50	CFRP sheet	0.7 Pu	100	85.5 (57.2)	124.5 (83.3)	149.4	top flange buckling
CSB2	13	100	prestressed CFRP sheet	0.7 Pu	100	52.4 (55.2)	66.1 (69.6)	95	CFRP sheet rupture
CSB3	13	50	prestressed CFRP sheet	0.7 Pu	100	83.2 (54.8)	127 (83.7)	151.8	CFRP sheet rupture

## Data Availability

Some or all data, models, or code that support the findings of this study are available from the corresponding author upon reasonable request.

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
