# Peer review of "Flexural Behavior on Damaged Steel Beams Strengthened with CFRP Sheets Subjected to Overloading"

_polymers, 2022, doi:10.3390/polym14071419_

Round 1

Reviewer 1 Report

The submitted paper titled: “Flexural behavior on damaged steel beam strengthened with CFRP sheet subjected to overloading” presents results of test and numerical analyses of damaged steel beams strengthened with carbon-fiber reinforced polymer (CFRP) sheets subjected to overloading. Although paper fits well to the topics of the journal and it deals with a subject still open to question and high interest for the Engineering community, there are some important issues that should be amended and several questions that should be clarified. The following comments and suggestions are raised for Authors’ reference:

  • Further details and explanatory notes concerning the damaged levels of 50% and 100% loss of tension flange are required. In Fig. 1 it is mentioned a 50% and 100% loss of flange. How this artificial reduce the tension of the flange? How much is the length of the cut shown in Fig. 1?
  • In Fig.2 the way that the device applies prestress to CFRP sheet with the self-made stretching bed should be clarified. It is also not clear if the flanges have holes.
  • In Fig.3, the loading device of the test procedure and some terms such as Spreader Beam and Hydraulic Jack should be explained and described adequately.
  • In Fig.3 fill in the variable of P by the vector could be mentioned.
  • All load versus deflection curves should have the same x- and y-axes scale and maximum values for comparison reasons.
  • Some of the figures require significant improvement. Labels and Curves are difficult to be clarified and understood.
  • Figure 6 which displays the failure modes of the steel beams should be clarified and enriched with further details and explanatory notes concerning buckling of the top flange or sheet rupture. This figure is rather draft and unclear, without details in texting.
  • Verification of the finite element simulation is fairly described and established. How did the authors select the mesh size? Was there any parametric analysis conducted? The size of the finite elements should be mentioned. The influence of the finite element mesh on the accuracy of the numerical results should also be demonstrated. It is mentioned that finite element software does not “simulate” any material or structural element, but the software just provides the means to help the user proceed with any simulation. The user provides the input parameters as well the appropriate geometry, load, and type of finite elements as well boundary conditions. Thus, it is important when a study is performed using finite element software to provide all of the above. Then, experimental results are used to validate the material constitutive laws and all simulation input parameters.
  • The literature review provided is brief and rather shallow, lacking coherence and failing to establish the relevance of the work reported in the paper. The findings of recent and additional relative works should also be considered in order to further establish the research significance and to promote the objectives of this study.
  • Material and mechanical properties and geometric characteristics could be presented in an additional Table.
  • Some comments concerning the feasibility of the proposed self-made stretching bed testing methodology on the state of the practice would be useful.
  • Conclusions seem rather weak since they are not supported sufficiently by the results. More specific and sound concluding remarks should be reported concerning the findings of this work.

Reviewer 2 Report

The paper proposes a study on the Flexural behavior on damaged steel beam strengthened with 2 CFRP sheet subjected to overloading. The paper is clear and concise and it addresses an interesting and relevant problem. The English is acceptable. The experimental tests developed in the manuscript are a good contribution to the field, but the Authors should explain some of their choices. Finally, I suggest a major revision. In what follows, I list some comments and suggestions that can be addressed by the authors while finalizing the manuscript in a major revision process.

Figure 1 and 2 are unclear;
The dots in Figure 4, what do they indicate?
Figure 1  is unclear;
In chapter4, the Authors are invited to describe the numerical model in a more appronfident way. In particular the points:
Models of the materials used (add references);
Type of contact;
bond between two materials;
Bond law;
Type of analysis adopted.

Round 2

Reviewer 1 Report

The manuscript has not been revised according the comments, some responses to the points seems to be incomplete. For example, point 5, 6, 7, 8 and 11. Reconsider them and please rerevise the manuscript accordingly.  

Reviewer 2 Report

The manuscript can be published in the present form

Author Response

The reviewer said 'the manuscript can be published in the present form'.